# Molecular Mechanisms for the Regulation of Nuclear Membrane Integrity

**DOI:** 10.3390/ijms242015497

**Published:** 2023-10-23

**Authors:** Ga-Eun Lee, Jiin Byun, Cheol-Jung Lee, Yong-Yeon Cho

**Affiliations:** 1BK21-4th, and BRL, College of Pharmacy, The Catholic University of Korea, 43, Jibong-ro, Wonmi-gu, Bucheon-si 14662, Gyeonggi-do, Republic of Korea; rkeoddl520@naver.com (G.-E.L.); wlssl_222@naver.com (J.B.); 2Research Center for Materials Analysis, Korea Basic Science Institute, 169-148, Gwahak-ro, Yuseong-gu, Daejeon 34133, Chungcheongnam-do, Republic of Korea; 3RCD Control and Material Research Institute, The Catholic University of Korea, 43, Jibong-ro, Wonmi-gu, Bucheon-si 14662, Gyeonggi-do, Republic of Korea

**Keywords:** inner nuclear membrane, type II membrane protein transcription factor, integral proteins in INM, INM integrity, nuclear morphology

## Abstract

The nuclear membrane serves a critical role in protecting the contents of the nucleus and facilitating material and signal exchange between the nucleus and cytoplasm. While extensive research has been dedicated to topics such as nuclear membrane assembly and disassembly during cell division, as well as interactions between nuclear transmembrane proteins and both nucleoskeletal and cytoskeletal components, there has been comparatively less emphasis on exploring the regulation of nuclear morphology through nuclear membrane integrity. In particular, the role of type II integral proteins, which also function as transcription factors, within the nuclear membrane remains an area of research that is yet to be fully explored. The integrity of the nuclear membrane is pivotal not only during cell division but also in the regulation of gene expression and the communication between the nucleus and cytoplasm. Importantly, it plays a significant role in the development of various diseases. This review paper seeks to illuminate the biomolecules responsible for maintaining the integrity of the nuclear membrane. It will delve into the mechanisms that influence nuclear membrane integrity and provide insights into the role of type II membrane protein transcription factors in this context. Understanding these aspects is of utmost importance, as it can offer valuable insights into the intricate processes governing nuclear membrane integrity. Such insights have broad-reaching implications for cellular function and our understanding of disease pathogenesis.

## 1. Introduction

The nuclear envelope (NE), composed of inner and outer nuclear membranes, plays a vital role in maintaining the integrity of genetic material and shielding it from various harmful molecules. It also provides structural support to the nucleus, primarily mediated by intermediate filament proteins like lamins located within the inner nuclear membrane (INM) [1]. The INM contains specific integral membrane proteins that contribute to its functions. The outer nuclear membrane (ONM) is continuous with the endoplasmic reticulum (ER) and houses ribosomes, while the INM is specialized for nuclear functions. Dysregulation of the integrity of the inner nuclear membrane, often caused by mutations or alterations in genes encoding INM-associated proteins, has been linked to several human diseases [2]. These diseases encompass a wide range of conditions, including muscular dystrophies, premature aging syndromes, lipodystrophies, cardiomyopathies, cancer, neurological disorders, and neuromuscular disease [3,4]. The commonality among these diseases is the disruption of NE structure and function, which significantly influences cellular homeostasis and contributes to tissue-specific pathologies. Despite the pivotal role played by the NE in determining cellular fate, the regulation of nuclear membrane integrity has not been comprehensively studied until now. This review addresses the molecular mechanisms and key molecules involved in regulating inner nuclear membrane integrity. Furthermore, it explores the potential role of type II integral proteins located at the nuclear inner membrane. Understanding these regulatory mechanisms is crucial for gaining insights into various disease processes and cellular homeostasis.

## 2. Structure of Nuclear Membrane

The NE comprises two closely juxtaposed lipid bilayers, forming a double-membrane structure known as the INM and the ONM. It encloses the nucleus in eukaryotic cells and consists of several distinct components, including the outer and inner membranes, nuclear pore complexes, nuclear lamina, and perinuclear space [5]. This complex structure serves as a critical barrier and regulator of molecular traffic between the nucleus and the cytoplasm, facilitating the passage of molecules in and out of the nucleus [6].

Beyond its role in molecular traffic, the NE acts as a physical barrier that segregates the genetic material (DNA) within the nucleus from the cytoplasm of the cell [7,8]. Additionally, it plays a vital role in gene regulation and nuclear organization, making it a fundamental component of eukaryotic cell biology [7,8].

The NE not only controls molecular movement but also provides mechanical support and helps maintain the structural integrity of the nucleus. This mechanical support is achieved through the interconnection of cytoskeletal elements on its nuclear and cytoplasmic sides, forming an intricate membrane–protein–chromatin network [9,10,11]. Key proteins found within the inner nuclear membrane, including lamins, various nuclear pore complex components, nucleoskeletal and cytoskeletal proteins (collectively known as the LINC complex), lamin B receptor (LBR), lamina-associated polypeptide (LAP) 1, LAP2, emerin, and MAN1, are well-known for their roles in preserving nuclear morphology [12,13,14]. Figure 1 summarizes the structure of NE and constituents contributing nuclear morphology maintenance.

Moreover, it is now widely recognized that the NE and its associated proteins have crucial roles in a range of cellular processes. These processes encompass cell division, cell signaling, transcription, cell cycle progression, chromosome tethering, cytoplasmic–nuclear transport, and cell migration [10,15,16]. However, the molecular mechanisms governing nuclear membrane integrity during apoptosis and necroptosis remain an area of active research, with much to be elucidated.

### 2.1. Outer Nuclear Membrane

The outer nuclear membrane is continuous with the ER membrane, forming a physical connection due to their shared phospholipid bilayer. This connection allows for the exchange of lipids and proteins and enables the ER to play a role in the synthesis and transport of these molecules. Maintaining the shape of the nucleus relies on a protein meshwork called the nuclear lamina. While the nuclear lamina primarily associates with the inner nuclear membrane, the outer nuclear membrane indirectly contributes to nuclear shape maintenance. It does so through interactions with the nuclear lamina, thereby reinforcing the structural integrity of the NE [17,18,19].

The nuclear lamina, situated just beneath the inner nuclear membrane, acts as a scaffold, providing mechanical support to the nucleus. It achieves this by interacting with the inner nuclear membrane, genomic DNA (chromatin), and nuclear pore complexes [6,11]. These interactions occur through integral membrane proteins known as LINC complexes. LINC complexes span both the inner and outer nuclear membranes, enabling the transmission of pulling forces from the nuclear lamina to the cytoskeleton in the cell’s cytoplasm [20,21,22]. This interplay between the nuclear lamina, LINC complexes, and the cytoskeleton ensures the mechanical stability of the nucleus. Such stability is essential for preserving the overall shape of the nucleus, preventing deformation or collapse in response to mechanical forces within the cell [14,22].

### 2.2. Inner Nuclear Membrane

The INM houses various integral proteins categorized into distinct groups. These include proteins permanently embedded in the INM using single, double, or multiple transmembrane domains, such as emerin, SUNs, LEM, and LBR. Additionally, there are INM-anchoring farnesylated proteins like B-type lamins and INM-interacting peripheral proteins like A-type lamins [23,24]. One critical component of the INM is the trimeric Sad1/UNC-84 (SUN) domain-containing SUN1/2 proteins. These proteins span the INM and bind to the tails of nesprins in the perinuclear space, forming the LINC complex [14,21,25]. Nesprin 1/2, found in the ONM, interacts with actin in the cytoplasm, while nesprin 3 interacts with intermediate filaments through plectin or microtubules through Dynein [26]. The LINC complex facilitates an outward pulling force via the cytoskeleton, helping to maintain the round nuclear shape against the expansive force of densely packed genomic DNA [13,14,27]. Another group of INM proteins, exclusively localized within the INM, remains relatively uncharacterized. This group, comprising approximately 60 integral membrane proteins referred to as NE transmembrane proteins or NETs, includes LBR, LAP1, LAP2, emerin, and MAN1. Some of these INM proteins interact with lamins and chromatin, indicating their involvement in various nuclear functions, including chromatin organization, gene expression, and DNA metabolism [28,29,30].

The significance of INM proteins is underscored by their link to numerous human diseases, sparking substantial interest in the field of NE biology. Improper localization and function of INM proteins have been associated with a range of disorders [20]. Notably, the LINC complex, particularly through SUN proteins, plays a role in influencing chromatin organization. SUN proteins have been implicated in tethering chromatin to the NE, potentially contributing to the spatial arrangement of the genome and influencing gene expression and regulation [10,31,32]. It is important to emphasize that the precise mechanisms underlying the influence of the LINC complex on genomic DNA and gene expression are ongoing areas of research. The interplay between nuclear positioning, chromatin organization, and gene regulation is intricate and context-dependent [14]. Nonetheless, the ability of the LINC complex to bridge the NE and cytoskeleton positions it as a crucial player in cellular processes that ultimately impact genomic function and gene expression [14,21,33]. In this section, we provide a summary of the characteristics and roles of integral membrane proteins within the INM.

## 3. Equilibrium of the Physical Force for Nuclear Morphology

The tension of the nuclear membrane, comprising both the inner and outer nuclear membranes, is subject to intricate cellular regulation. Precise control of nuclear membrane tension is vital for maintaining the structural integrity of the nucleus and for governing essential cellular processes, including nuclear positioning, cell division, and gene expression [34,35]. Several mechanisms regulate both inward and outward tension of the nuclear membrane. One central player in this regulation is the Linker of the LINC complex. Comprising proteins like SUN and Klarsicht, ANC-1, and Syne homology (KASH) domain proteins, the LINC complex connects the nuclear lamina on the inner nuclear membrane to the cytoskeleton on the outer nuclear membrane [36,37]. This connection allows for the transmission of forces from the cytoskeleton to the nuclear membrane, thereby regulating tension. SUN proteins, for example, interact directly with the C-terminal tail domain of Lamin A and Lamin C, influencing genomic DNA by tethering chromatin regions to the inner nuclear membrane, contributing to the spatial organization of the genome [32,37,38].

The actin cytoskeleton, composed of actin filaments, can exert forces on the nuclear membrane through actin-binding proteins and motor proteins like myosins [39]. These proteins interact with the NE and generate tension by pulling on the outer nuclear membrane [8,40]. The regulation of this tension involves controlling actin polymerization and motor protein activity. Microtubules, another component of the cytoskeleton, also influence nuclear membrane tension. Microtubule-associated motor proteins, such as dynein and kinesin, interact with the NE. Dynein exerts an inward force on the nucleus, while kinesin generates outward forces. The dynamic assembly and disassembly of microtubules play a role in regulating these tensions [26,41].

The nuclear lamina, a protein meshwork lining the inner nuclear membrane, provides structural support to the nucleus and contributes to tension regulation. Modifications of lamin proteins, such as lamin A and lamin B, through processes like phosphorylation, can alter their interactions with chromatin and the cytoskeleton, influencing tension [19,23,24].

The maintenance of nuclear membrane tension is critical because the tightly packed genomic DNA within the nucleus exerts an inward pressure while the cytoskeleton in the cytoplasm applies an outward pulling force [42]. The balance between these forces is essential for preserving nuclear morphology. However, in Progeria syndrome (Hutchinson–Gilford Progeria Syndrome, HGPS), one of the key cellular abnormalities is the loss of nuclear membrane tension [8,43,44,45]. This loss of tension is associated with the accumulation of an abnormal protein called progerin, resulting from a mutation in the LMNA gene [43,44]. Progerin accumulates in the nucleus and disrupts the normal structure and function of the NE [45,46]. This disruption leads to an irregular and distorted NE, a loss of structural stability, impaired gene regulation, DNA repair processes, and contributes to the premature aging phenotype observed in Progeria. It manifests in various characteristic symptoms of the syndrome. We summarize how physical forces can contribute to nuclear morphology maintenance in Figure 2.

## 4. The Roles of Constituents in the NE Involved in Nuclear Morphology Maintenance by Complex Formation

The ONM and INM are connected at nuclear pore complexes and form the NE. Integral proteins in the INM play crucial roles in maintaining the structure and function of the nucleus, including regulating nuclear–cytoplasmic transport, and connecting the nuclear lamina to the cytoskeleton, contributing to various cellular processes and functions. Table 1 summarizes the role of each constituent in the NE involved in the nuclear morphology maintenance by complex formation.

### 4.1. Constituents Tethered in the INM

#### 4.1.1. Lamins

Lamins are a group of proteins that constitute a fundamental component of the nuclear lamina, a network of filaments situated within the nucleus of eukaryotic cells. The nuclear lamina serves to provide structural support to the nucleus and plays a pivotal role in various nuclear functions, including DNA replication, transcription, and the maintenance of nuclear shape [24,47]. Two primary types of lamins exist: A-type lamins, represented by lamin A and lamin C, and B-type lamins, which encompass lamin B1 and lamin B2. Lamin A, a major constituent of the nuclear lamina, is initially synthesized as prelamin A [24,48,49]. This precursor undergoes several post-translational modifications, including farnesylation and cleavage, to attain its mature lamin A form. Lamin C, a splice variant of the LMNA gene, shares structural similarities with lamin A [50]. Mutations in the LMNA gene are associated with a group of genetic disorders known as laminopathies. Notable examples include Hutchinson–Gilford Progeria syndrome (HGPS), characterized by accelerated aging, and Emery–Dreifuss muscular dystrophy [44,51].

Lamin B1, another significant component of the nuclear lamina, is indispensable for preserving the structural integrity of the NE. It interacts with chromatin and contributes to genome organization within the nucleus [17,47]. Mutations in the LMNB1 gene can result in conditions such as autosomal dominant Emery–Dreifuss muscular dystrophy and certain leukodystrophies, primarily affecting muscle and nervous tissues [52]. Lamin B2, also classified as a B-type lamin, plays a similar role to lamin B1. It contributes to NE integrity, chromatin organization, and overall nuclear stability. Mutations in the LMNB2 gene can lead to diseases affecting the NE and cellular function [19,53,54].

The interaction between lamins and chromatin plays a crucial role in maintaining nuclear membrane integrity. This interaction helps provide structural stability to the nucleus and regulates various nuclear processes [55]. Since the chromatin is anchored to the INM via lamins, this attachment contributes mechanical forces evenly across the nucleus and prevents excessive deformation of the nuclear envelope during cellular activities [56]. In summary, lamins are crucial for providing mechanical stability to the nucleus and maintaining its morphology.

#### 4.1.2. Chromatin

The terms heterochromatin and euchromatin refer to two distinct structural states of chromatin, the complex of DNA and proteins found in the nucleus of eukaryotic cells [57,58]. Heterochromatin is a tightly packed form of chromatin that is transcriptionally inactive [59,60]. It consists of highly condensed DNA, making it less accessible to the transcriptional machinery [60]. Heterochromatin is associated with regions of the genome that are typically silenced or not actively transcribed, appearing as dark, dense regions under a microscope [61,62,63]. In contrast, euchromatin is a loosely packed form of chromatin that is transcriptionally active [57,61,64]. It contains open and accessible DNA, making it available for transcription by RNA polymerase and other transcription factors [61,65]. Euchromatin appears as a more extended and less dense region under a microscope [66].

The INM is one of the two lipid bilayers that make up the NE, the boundary separating the nucleus from the cytoplasm [5,10,67]. The organization of chromatin within the nucleus is not uniform, and heterochromatin and euchromatin can be found at specific locations within the nuclear space [5,67]. Heterochromatin is often associated with the nuclear periphery, including the inner nuclear membrane [16]. This association is important for maintaining the stability and integrity of the genome [68,69]. Heterochromatin at the INM can be involved in functions such as gene silencing, DNA repair, and chromosome organization [69,70].

Euchromatin is generally found more centrally within the nucleus, away from the nuclear periphery [63,69,71]. This positioning allows active genes to be closer to the transcriptional machinery, facilitating gene expression [57,72]. While euchromatin may not be directly associated with the INM, it can interact with the nuclear lamina, a protein meshwork lining the inner nuclear membrane, which plays a role in regulating gene expression [18,67,73].

Heterochromatin and euchromatin play significant roles in maintaining nuclear morphology, primarily through their impact on the organization and structural components of the nucleus [74,75]. Heterochromatin influences nuclear shape and compaction, typically associated with regions of tightly condensed DNA [57,61,76]. Its presence at the nuclear periphery and interaction with the nuclear lamina help maintain the overall shape and structural integrity of the nucleus [77]. Heterochromatin compaction contributes to the flattening or indentation of the NE, which can affect nuclear morphology [74,78]. In cases of laminopathies or mutations in nuclear lamina proteins, abnormal heterochromatin organization can lead to alterations in nuclear shape, contributing to diseases characterized by NE defects [79]. Moreover, during mechanical stress, such as cell migration or changes in cell shape [80], the interaction between lamins and chromatin helps protect the nuclear envelope [81]. It prevents excessive stretching or rupture of the envelope, which could otherwise lead to DNA damage and cellular dysfunction and regulates gene expression [81].

Heterochromatin also plays a crucial role in genome stability by silencing repetitive DNA elements like transposons and satellite repeats, thus preventing genome instability and aberrant recombination events [82]. This stability contributes to the overall structural integrity of the nucleus. Heterochromatin can interact with the nuclear lamina, a protein meshwork lining the inner nuclear membrane, anchoring heterochromatic regions to the NE [49,83]. This interaction is essential for maintaining the structural integrity of the NE [56].

In contrast to heterochromatin, euchromatin, being transcriptionally active, contains genes that need to be transcribed [57,84]. Typically, euchromatin is positioned closer to nuclear pores, which are openings in the NE for the export and import of molecules. This proximity facilitates the efficient export of mRNA and other gene products, contributing to nuclear morphology by ensuring the flow of genetic information between the nucleus and cytoplasm. This flow of genetic information regulates gene expression involved in NE remodeling and cytoskeletal dynamics, resulting in dynamic changes in nuclear shape depending on the cellular context [85]. Therefore, these two chromatin states, heterochromatin and euchromatin, play critical roles in genome organization, gene regulation, and overall cellular function. Their organization and regulation are essential for maintaining cell viability and integrity [86]. Therefore, the interaction between chromatin and lamins is a fundamental aspect of nuclear biology that contributes to nuclear membrane integrity. This interaction provides structural stability to the nucleus [47,55].

#### 4.1.3. Lamina-Associated Domains (LADs)

Lamina-Associated Domains (LADs) are specific genomic regions in the nucleus of eukaryotic cells that are in close proximity to the INM and interact with the nuclear lamina [73,87]. LADs play a crucial role in nuclear architecture, gene regulation, and genome organization [87]. LADs are composed of several components that collectively contribute to their structure and function within the cell nucleus [73]. The primary structural component of LADs is the nuclear lamina, which consists of lamin proteins [73,87,88]. Lamins are intermediate filament proteins that form a meshwork on the inner surface of the INM [24]. LADs are also composed of chromatin, which includes DNA and associated histone proteins (dimethylated lysine 9 of histone H3) [87,89]. In particular, LADs are enriched in heterochromatin, a densely packed and transcriptionally inactive form of chromatin [73]. The presence of heterochromatin in LADs contributes to gene silencing and a repressive chromatin environment [87,90]. Various protein complexes and factors are associated with LADs to mediate their function. These may include chromatin-remodeling complexes, transcriptional repressors (such as non-coding RNAs), and other regulatory proteins [91]. These complexes assist in maintaining the repressive environment of LADs and facilitate interactions with other nuclear structures [91]. LADs often have boundary elements or insulator sequences at their edges. Boundary elements or insulator sequences, DNA sequences and associated proteins, demarcate the boundaries of LADs and prevent the spread of heterochromatin into adjacent euchromatic regions and can regulate gene expression by acting as insulators [73]. CCCTC-binding factor (CTCF) is a highly conserved DNA-binding protein that plays a central role in chromatin organization [73,92,93]. It acts as an insulator by binding to specific DNA sequences known as CTCF-binding sites or motifs [73,93]. The canonical CTCF-binding motif is characterized by a 20-to-22-base-pair sequence that is rich in CpG dinucleotides and exhibits a specific pattern of sequence conservation [94]. The core sequence recognized by CTCF is often described as 5′-CC(N6)GG-3′. When CTCF binds to these sites, it can create a physical barrier that prevents the spread of heterochromatin and the formation of LADs or other repressive chromatin structures [94,95]. Boundary elements are often enriched with CTCF-binding sites. Boundary elements play a crucial role in maintaining the three-dimensional organization of the genome [96]. Moreover, CTCF mediates long-range chromatin interactions through the assistance of cohesion and other proteins, resulting in the formation of chromatin loops [73,94,95]. By tethering distant genomic regions together, CTCF-mediated looping can facilitate gene regulation, insulate regions from the spread of heterochromatin, and contribute to the organization of LADs and other chromatin domains [97,98]. Frequently, insulator elements may be marked by histone H3 lysine 4 (H3K4) methylation, which is typically associated with transcriptionally active chromatin [99]. The involvement of the LAD in the nuclear membrane integrity is caused by which dysregulation of LAD interactions can be associated with various diseases, including laminopathies [73]. Although the mutations at the LAD have not been identified, aberrant interaction by lamins induces large-scale disruptions in chromatin organization and nuclear shape [73,87]. Therefore, without proper LAD interactions, the nuclear envelope might become more fragile and prone to damage.

#### 4.1.4. Barrier-to-Autointegration Factor (BAF)

BAF, Barrier-to-Autointegration Factor, is a protein that plays a crucial role in maintaining the structural integrity and organization of the cell nucleus [100]. It is primarily associated with the INM and has several important functions, including NE integrity, chromatin organization, DNA replication and repair, nuclear lamina interaction, and cell division [101,102]. BAF interaction with lamins assists in anchoring chromatin to the inner nuclear membrane, which is essential for maintaining the structural integrity of the NE [102,103]. This interaction stabilizes the NE and contributes to its overall architecture [102,103,104]. BAF directly interacts with lamins, particularly lamin A/C [105]. This interaction is mediated through N-terminal domain of BAF and C-terminal tail domain of lamin A/C [105,106]. Moreover, the BAF protein contains a DNA-binding domain. Both domains of BAF are involved in binding to lamins [106]. This direct binding between BAF and lamin A/C is a key component of their interaction [106,107].

BAF has DNA-binding activity. It can directly bind to DNA sequences, including AT-rich regions [100,104,108]. When BAF binds to DNA, it can induce a conformational change in the DNA molecule, leading to compaction [109]. This property is important in various cellular processes, such as chromatin organization and the formation of higher-order chromatin structures [108,109]. By binding to specific DNA sequences within the chromatin, BAF plays a role in tethering chromatin to the INM by binding to specific DNA sequences within the chromatin [109,110]. This tethering is important for maintaining the structural integrity of the NE and organizing chromatin into distinct nuclear domains [102,109]. BAF is involved in organizing and positioning chromatin within the nucleus. It contributes to the formation of distinct nuclear domains and the regulation of gene expression by anchoring specific genomic regions to the INM [97,104]. BAF plays a role in DNA replication and repair processes by tethering DNA to the NE [111]. This facilitates proper DNA replication and ensures efficient DNA damage response and repair [109,111]. During cell division, BAF is involved in the reassembly of the NE around daughter nuclei [103,112,113]. It aids in the proper segregation of chromosomes and the reformation of the NE after mitosis [113]. Therefore, the N-terminal domain of BAF and the C-terminal tail domain of lamin A/C are key players in this interaction, which is essential for maintaining nuclear envelope integrity, chromatin organization, and other nuclear functions [114].

#### 4.1.5. Heterochromatin Protein 1 (HP1)

Heterochromatin Protein 1 (HP1) is a pivotal contributor to the establishment and upkeep of heterochromatin, a densely packed and transcriptionally silent genomic region in eukaryotes [115,116,117]. HP1′s engagement with heterochromatin is governed by a complex interplay of molecular mechanisms encompassing protein–protein interactions, post-translational modifications, and alterations in chromatin structure [115,116,118]. Central to HP1′s function is its highly conserved chromodomain, which specifically recognizes and binds to histone H3 trimethylated at lysine 9 (H3K9me3), a characteristic hallmark of heterochromatin [119,120,121]. The chromodomain’s ability to create a stable complex with H3K9me3 relies on hydrophobic interactions and hydrogen bonding [120,122]. HP1 has the capacity to form homodimers or heterodimers due to a dimerization interface within its chromodomain, enabling the establishment of an intricate network of HP1 proteins within heterochromatin [121,122,123]. This network enhances the compaction and stability of the heterochromatin structure [124]. HP1 further collaborates with various proteins associated with heterochromatin assembly and maintenance [125]. For instance, its interaction with HP2 or heterochromatin-associated proteins, including HP1-interacting proteins and Swi6/HP1-interacting protein, reinforces and modulates heterochromatin structure and function, respectively [125,126,127]. Notably, post-translational modifications of HP1, such as phosphorylation and methylation, play a pivotal role in regulating its chromatin binding [124,128,129]. Phosphorylation of HP1α at S10 by Aurora B kinase weakens its association with H3K9me3, resulting in its dissociation from chromatin and subsequent transcriptional activation at target loci [130]. Moreover, HP1α phosphorylation at Ser51 by protein kinase CK2 is implicated in gene derepression and correlates with reduced levels of H3K9me3 [131]. Additionally, HP1, by virtue of its binding to H3K9me3, recruits other chromatin-modifying enzymes like histone methyltransferases and histone deacetylases to heterochromatin, thereby promoting chromatin condensation and gene expression repression [115,118,132]. Notably, once HP1 binds to H3K9me3, it can propagate along the chromatin fiber, facilitating the extension of heterochromatin formation across an extended genomic region [133]. Therefore, these interactions and modifications contribute to the compaction and transcriptional silencing of heterochromatin, ultimately maintaining genome stability, regulating gene expression, and nuclear membrane integrity in eukaryotic cells.

### 4.2. Integral Membrane Proteins in the INM

Integral proteins in the INM play crucial roles in maintaining the structure and function of the nucleus. The NE consists of two membranes: the ONM and the INM, which are connected at nuclear pore complexes [134,135]. While the ONM is continuous with the ER and contains ribosomes, the INM is specialized for nuclear functions and contains specific integral membrane proteins [136].

#### 4.2.1. SUN

SUN proteins are a family of integral membrane proteins that are primarily found in the INM of the NE [32]. They play a crucial role in various cellular processes, including nuclear positioning, nuclear migration, meiosis, and maintaining the structural integrity of the nucleus [32,137].

SUN proteins typically have a conserved domain structure that includes a single transmembrane domain (TMD) near the C-terminus, anchoring them in the INM [138]. The N-terminal region of SUN proteins contains the SUN domain, which is essential for their interactions with other proteins, including KASH domain-containing proteins in the ONM [13,139,140]. SUN proteins are often classified into different subtypes based on the presence of additional domains or variations in their sequence. The primary function of SUN proteins is to interact with KASH domain-containing proteins in the ONM [140,141]. KASH proteins extend into the perinuclear space and connect with the cytoskeleton, including microtubules and actin filaments [138]. The interaction between SUN and KASH proteins forms a bridge known as the LINC complex [22,138]. This complex spans the NE and connects the nucleoskeleton (formed by lamins and other nuclear components) to the cytoskeleton, allowing for the transmission of mechanical forces and positional information between the nucleus and the cytoplasm [14,21,142].

The LINC complex, facilitated by SUN-KASH interactions, plays a crucial role in nuclear positioning within the cell [138]. It helps to anchor the nucleus in a specific location within the cell, which is essential for various cellular processes, including cell migration, division, and differentiation [22,143]. In migrating cells, the LINC complex is involved in positioning the nucleus at the cell’s rear to allow for efficient migration [143]. SUN proteins are involved in nuclear migration during various developmental processes, including meiosis in certain cell types [144]. They help guide the movement of the nucleus to specific cellular locations, which is vital for proper cell function and development [145]. SUN proteins, through their connections with the nuclear lamina, contribute to maintaining the shape and size of the nucleus [25,146]. Dysregulation can lead to abnormal nuclear morphology [25,147]. Mutations or dysregulation of SUN proteins can lead to various diseases and developmental disorders [147]. For example, mutations in the SUN1 gene have been associated with meiotic defects and male infertility [148]. Aberrant LINC complex function can also contribute to NE-related diseases, such as muscular dystrophies and Progeria [149]. Therefore, the LINC complex-mediated connection between the INM and ONM and the cytoskeleton provides structural stability to the nucleus, facilitates nuclear positioning, regulates mechanotransduction, and maintains the composition and function of the nuclear envelope and its associated components [21].

#### 4.2.2. Emerin

Emerin is a key component of the NE [150,151]. Therefore, emerin participates as a component of the nuclear lamina [150,151]. Emerin, a protein found in the inner nuclear membrane of the cell’s nucleus, plays several important roles in maintaining nuclear structure and function [152,153]. Emerin involves maintaining the structural integrity of the NE by interacting with lamins and other inner nuclear membrane proteins [153]. Emerin is involved in regulating chromatin organization within the nucleus [154,155]. It interacts with chromatin-associated proteins and helps anchor chromatin to the nuclear periphery [154]. This anchoring influences gene expression by modulating the accessibility of specific regions of DNA [156,157]. Emerin has been shown to interact with transcription factors, such as GATA1 (involved in hematopoiesis), LDB1 (involved in the regulation of muscle-specific genes), and MyoD (involved in muscle development and differentiation) [158], and other regulatory proteins, such as lamin A/C and Bcl-2 [153], suggesting that it may play a role in gene expression regulation [150,151,157]. It can influence the localization of transcription factors within the nucleus, potentially affecting their ability to activate or repress specific genes [151,156,157]. Emerin interacts directly with components of the cytoskeleton, such as actin filaments, via a conserved actin-binding domain by protein–protein interaction and indirectly by the formation of the LINC complex, resulting in the participation of various cellular processes, including nuclear positioning and migration [14,22,159]. Emerin’s role in linking the nucleus to the cytoskeleton helps coordinate cellular responses to mechanical forces and signaling cues [152,159]. Since emerin interacts with actin filaments through direct binding and by forming a bridge with linker proteins like nesprins, emerin interaction as a part of the LINC complex plays a critical role in connecting the nucleus to the cytoskeleton, facilitating nuclear movement, gene regulation, and signaling in response to mechanical cues via the nuclear membrane integrity [160].

#### 4.2.3. Lamin B Receptor (LBR)

Some studies suggest that lamin proteins may interact directly with specific DNA sequences within LADs [73,87,161]. These sequences are often referred to as “lamin-associated motifs” or “Lamin B receptor (LBR) recognition motifs”, generally consisting of a DNA sequence rich in A and T [73,90]. LBR is a transmembrane protein found in the inner nuclear membrane of the cell nucleus, and it plays a role in anchoring chromatin to the NE [161,162,163]. LBR is a protein that interacts with lamin B1 and is involved in NE assembly [161,164]. The binding of lamin proteins to these motifs within LADs may help anchor chromatin to the nuclear periphery [73,87,90].

The LBR is a transmembrane protein found in the INM of the cell nucleus, and it is involved in anchoring chromatin to the NE [161,162]. While LBR does interact with DNA, it does not have a known sequence-specific DNA-binding domain like transcription factors [165]. Instead, LBR’s DNA interactions are mediated through other proteins, such as heterochromatin-associated proteins and lamina-associated factors [29,166,167].

LBR is believed to interact with specific chromatin regions, including LADs [29,166,167]. These interactions are thought to occur through protein–protein interactions rather than direct DNA binding [161,167]. LBR may indirectly associate with specific DNA sequences through its interactions with other nuclear components. The exact molecular details of how LBR associates with chromatin and the specific sequences involved are still an active area of research. The association between LBR and chromatin likely involves a combination of protein–protein interactions with other NE proteins (such as emerin and lamins) and potentially interactions with chromatin-associated factors [29,168]. It is important to note that LBR’s role in anchoring chromatin to the NE contributes to the spatial organization of the genome, impacting gene expression and genome architecture [38,161]. However, the exact mechanisms and sequences involved in these interactions are complex and not fully understood.

#### 4.2.4. MAN1

Membrane-anchored Nucleus Protein 1 (MAN1), a protein found in the inner nuclear membrane (INM), plays pivotal roles in upholding the integrity of the INM and the overall nuclear envelope structure [169]. MAN1 is among the integral membrane proteins that contribute to anchoring the nuclear envelope to the nuclear lamina, thus preserving the structural robustness of the nuclear envelope [169]. The interaction between MAN1 and lamins, specifically lamin A and B, provides critical support to the nucleus, preventing the collapse or deformation of the nuclear envelope [170]. The LAP2-Emerin-MAN1 (LEM) domain of MAN1 is instrumental in its interaction with lamin proteins [170]. The LEM domain encompasses a distinctive structural motif that facilitates its binding to the C-terminal tail domains of lamins, particularly lamin A/C [171]. The interaction between MAN1 and lamins is mediated by hydrophobic and electrostatic interactions involving Leu, Ile, Phe, Arg, and Lys residues within the MAN1 LEM domain and the hydrophobic regions of lamin A/C [171]. Furthermore, the interaction between MAN1 and lamins frequently contributes to the formation of the LINC complex [147,172]. Consequently, this interaction furnishes essential structural support to the nucleus, aiding in the maintenance of nuclear shape and preventing deformation of the nuclear envelope during various cellular processes, including cell movement, nuclear positioning, and mechanical stresses. Despite its critical role in regulating nuclear membrane integrity, the precise molecular mechanisms underlying MAN1′s involvement in nuclear membrane integrity have not yet been fully elucidated.

#### 4.2.5. Transcription Factors in the INM

Type II membrane proteins are a class of integral membrane proteins that have a single transmembrane domain and are anchored to the membrane by this domain [173,174]. These proteins typically span the lipid bilayer once, with one end (the N-terminus) facing the cytoplasm and the other end (the C-terminus) facing the extracellular space or an organelle lumen [173,175]. In some cases, type II membrane proteins can contain transcription factor activity or be associated with transcriptional regulation [176,177,178,179]. These proteins, including steroid hormone receptors (estrogen and androgen receptors), generally reside at the endoplasmic reticulum and are transported into the Golgi complex (Gc) by appropriate stimuli [180]. At the Gc, these proteins are cleaved by S1P and S2P proteases, resulting in the release and nuclear localization of the N-terminal domains, which contains DNA-binding motifs at the specific DNA sequences [177,178,181]. Some proteins, such as sterol regulatory element-binding proteins, Notch receptors, and activating transcription factor 6, are activated by similar processing mechanisms and act as transcription factors to respond depending on the cellular context [177,182]. Although Lamins and emerin are type II membrane proteins that can be localized into the INM and bound with the genomic DNA, these proteins do not act as transcription factors [183,184]. Recent results demonstrated that CREB3-L1, OASIS, can be localized into the INM, especially new synthesizing nuclear membrane [185]. However, CREB3-L1 acting as a transcription factor has not been elucidated.

When unfolded proteins accumulate in the ER, ATF6α is transported to the Gc, resulting in the release of an active cytoplasmic fragment of ATF6 (ATF6(N) by the S1P/S2P-mediated cleavage. The ATF6(N) then translocates to the nucleus, where it acts as a transcription factor. In the nucleus, ATF6(N) binds to specific DNA sequences known as ER stress response elements (ERSEs) or ER stress response elements II (ERSE-II) [186,187]. It then activates the transcription of target genes that play a role in alleviating ER stress, such as chaperones and proteins involved in ER-associated degradation (ERAD) [188]. However, immunocytofluorescence analysis of ATF6 using ATF6-GFP clearly showed that a strong green fluorescence belt is detected surrounding the nucleus [177,189,190]. This fluorescence belt is not overlapped with GM130, a Gc marker [189], and PDI, a ER marker [190]. Similar observation by immunocytofluorescence assay using Flag-CREB3-L1 full length showed that CREB3-L1 accumulates locally to the nuclear membrane [185]. Although all of these reports showed that the full length of ATF6 and CREB3-L1 shows the ER localization, the notable nuclear localization of ATF6 and CREB3-L1 suggests that there are unrevealed functions at the nuclear membrane. Since CREB3 and CREB3 isotypes, including CREB3-L1, -L2, -L3, and -L4, are type II membrane proteins harboring similar mechanisms with ATF6 for the activation [179,181], studies on the molecular mechanisms of these proteins for the nuclear membrane localization and roles at the nuclear inner membrane are necessary.

## 5. Nuclear Membrane Breakdown and Nuclear Morphology Change

Regulated cell death processes are characterized by distinctive alterations in nuclear morphology that are tightly controlled and have functional consequences. These changes not only serve as diagnostic markers but also have functional implications. Alterations in nuclear morphology can influence the clearance of dying cells, the release of inflammatory signals, and the immune response [8,191]. Understanding these nuclear morphology changes is essential for elucidating the mechanisms and consequences of different forms of regulated cell death in various physiological and pathological contexts [8]. Since nuclear morphology change is triggered by the disruption of tension equilibrium as mentioned in the introduction of this review, we now address the potential mechanism to induce the nuclear morphology change via nuclear membrane destabilization processes.

### 5.1. Proteases Cleaving SUN and KASH Interaction

The precise identity of the proteases responsible for cleaving SUN and KASH proteins localized within the INM remains elusive [37,143]. The existing literature suggests that various proteases become activated and participate in cleaving these proteins, contingent upon contextual factors such as cellular stressors, developmental stages, or specific cellular signaling pathways [178,192]. Numerous proteases have been implicated in the cleavage of components within the LINC complex.

Caspases, a family of cysteine proteases renowned for their pivotal role in apoptosis (programmed cell death), exhibit the ability to cleave a spectrum of NE proteins [8], including SUN and KASH proteins. The molecular basis upon which SUN and KASH proteins may be cleaved by Caspase-3 during apoptosis encompasses specific recognition motifs, proteolytic cleavage events, and consequential functional alterations. The identification of putative caspase cleavage sites within SUN and KASH proteins relies on the presence of the consensus sequence (Asp-X-X-X). Upon activation, Caspase-3 translocates to the nucleus, where it encounters its nuclear substrates [193,194], including SUN and KASH proteins. Caspase-3 cleaves these proteins at specific aspartic acid residues within the caspase recognition motifs, liberating N-terminal fragments from both SUN and KASH proteins. This cleavage may disrupt the interaction between SUN and KASH proteins, leading to the disassembly of the LINC complex. SUN protein cleavage releases their N-terminal SUN domains from the INM, while KASH protein cleavage may result in the release of their cytoplasmic tails. Consequently, this disassembly perturbs the normal maintenance of nuclear–cytoskeletal connections by the LINC complex, thereby altering nuclear morphology, including nuclear condensation and fragmentation. Additionally, it disrupts nuclear and envelope integrity, contributing to the orchestrated dismantling of cellular structures during programmed cell death (apoptosis).

Calpains, calcium-dependent proteases, exhibit the capability to cleave an array of cellular substrates, including components of the NE, such as SUN and KASH proteins [195]. Calpains recognize specific peptide sequences within their target proteins. The determination of calpain cleavage sites is contingent upon the presence of distinct amino acid motifs, typically characterized by hydrophobic or bulky residues at the P2 position (two amino acids preceding the scissile bond) and basic residues at the P1 position (the position immediately preceding the scissile bond). Calpain-mediated cleavage disrupts the interactions between SUN and KASH proteins, culminating in the disassembly of the LINC complex and subsequent loss of nuclear–cytoskeletal coupling. This disruption leads to abnormal nuclear morphologies, as evidenced during programmed cell death.

### 5.2. Nuclar Membrane Dynamicity and Nuclear Membrane Integrity

The stiffness and softness of the nuclear membrane play key roles in various cellular processes, including gene expression, nuclear transport, and cell division, and in determination of cell fate, including cell survival and death [1]. The stiffness of the nuclear membrane provides mechanical support to protect the genetic materials from mechanical stress and external forces. In contrast, the nuclear membrane also needs the softness to allow for nuclear envelope breakdown and reformation during cell division, such as mitosis and meiosis [196,197]. Since chromatin remodeling refers to the alteration of chromatin structure, which encompasses DNA and histones, to allow or restrict access to specific DNA regions for transcription, replication, repair, and other cellular activities, the INM integrity homeostasis in the dynamic nuclear environment is a key event to determine cell survival and death [85]. Mechanosensory mechanisms in cells detect changes in nuclear envelope stiffness or nuclear deformation and trigger signaling pathways that lead to chromatin structure alteration [198]. For example, mechanosensors activate chromatin-modifying enzymes that alter histone modifications or DNA methylation patterns, which are markers for heterochromatin or euchromatin [198]. Therefore, controlling the nuclear membrane stiffness and softness is linked to the contribution to various cellular and pathological processes. For example, excessive stiffness of the nuclear membrane led to a loss of nuclear envelope integrity, resulting in nuclear envelope ruptures, compromising the separation of nuclear and cytoplasmic contents, and potentially leading to DNA damage and cell death. Increased stiffness induces distortion of the nucleus, resulting in abnormal nuclear morphology, as shown in laminopathies [199]. In contrast, if the nuclear membrane becomes too soft, it loses its ability to maintain nuclear envelope integrity, resulting in not only nuclear envelope deformations, compromising the separation of nuclear and cytoplasmic contents, but also the positioning of chromatin within the nucleus, leading to errors in chromosome segregation and aneuploidy. Conditions like laminopathies, muscular dystrophies, and some types of cancer are associated with abnormalities in NE structure and mechanics [52,199,200].

### 5.3. Nuclear Membrane Breakdown and Nuclear Blebbing

Nuclear blebbing is the formation of membrane-bound, spherical protrusions or bulges from the nuclear envelope by the disassembly of nuclear lamina. The actin–myosin contractile forces generated by the cytoskeleton contribute to nuclear blebbing. In contrast, nuclear membrane invagination is a cellular process formed by the inward folding or involution of the nuclear envelope to create intranuclear membrane structures. In this process, endosomal sorting complexes required for transport (ESCRT) play a key role in nuclear membrane remodeling and invagination [201]. In particular, ESCRT-III subunits polymerize into spirals or filaments on the inner surface of the membrane, causing it to bend inward [201]. Eventually, the ESCRT machinery aids in the constriction and pinching off of the invaginated membrane structure, leading to its separation. This nuclear invagination occurs in cell division to segregate and protect the genetic material within the dividing cell and respond to DNA damage to isolate damaged DNA.

With parallel, the breakdown of the NE, commonly referred to as nuclear membrane breakdown or NE disassembly, represents a pivotal event in cell division (mitosis and meiosis) and is crucial in specific cellular processes, such as the reformation of the NE following mitosis [196]. This event facilitates the segregation and equitable distribution of genetic material to the daughter cells [202]. On the other hand, nuclear membrane rupture is a phenomenon that can transpire under diverse circumstances, encompassing specific cellular processes and pathological conditions [19,35]. This process entails the breach or disruption of the NE, which constitutes the double-membrane structure enveloping the cell’s nucleus [35,203]. Therefore, the difference between nuclear membrane breakdown and rupture is to be controlled or uncontrolled [7,204].

The molecular mechanisms of the nuclear membrane rupture and blebbing are main topics in apoptosis and necroptosis [8,205,206]. Apoptosis is a tightly regulated process that orchestrates the systematic dismantling of a cell, encompassing a sequence of events that culminate in the fragmentation and disintegration of the cell nucleus [8,207,208,209]. LBR plays a contributory role in this process by participating in the alteration of nuclear structure during apoptosis [30]. In the course of apoptosis, there is a profound transformation in the chromatin within the nucleus [210]. This transformation leads to the formation of densely compacted structures, commonly referred to as apoptotic bodies [211,212].

LBR plays a vital role in facilitating this condensation of chromatin by engaging with specific chromatin regions [161]. The perturbation of the NE and the reorganization of chromatin render DNA more susceptible to nucleases and other enzymes that mediate DNA cleavage throughout apoptosis [8]. The release of nuclear contents into the cytoplasm exposes DNA to the action of these enzymes. DNA cleavage during apoptosis is primarily orchestrated by endonucleases, including caspase-activated DNases (CADs) and endonuclease G (EndoG) [213,214]. For instance, in the course of apoptosis, caspase-3 and caspase-7 undergo activation through a series of proteolytic cleavage events initiated by the apoptotic signaling cascade [212,215,216]. Once activated, these caspases become enzymatically active [217]. Subsequently, caspase-3 and caspase-7 cleave an endogenous inhibitor of CAD, known as inhibitor of CAD (ICAD), also referred to as DNA fragmentation factor 45 kDa (DFF45) [214,215,218]. This cleavage event results in the dissociation of CAD from ICAD. The cleavage of ICAD by caspases reveals the active site of CAD [219]. The activated CAD then serves as a nuclease, relocating to the nucleus and cleaving chromosomal DNA at internucleosomal sites [220]. This cleavage process generates characteristic DNA fragments that are typically multiples of approximately 180–200 base pairs. These enzymes are activated during apoptosis and are responsible for the fragmentation of chromosomal DNA into smaller, discrete fragments [221]. The accessibility of chromatin to these enzymes is influenced by the structural modifications occurring within the nucleus, which encompass the breakdown of the nuclear envelope and the condensation of chromatin [8,222,223]. However, how molecular processes enable the materials to be encapsulated into the vesicle has not been clearly elucidated.

### 5.4. The Contents Originated from the Nucleus in Apoptotic Bodies

Apoptotic bodies are membrane-bound vesicles that contain various cellular components, including the contents of the nucleus [212,224]. The contents of the nucleus within apoptotic bodies reflect the dramatic changes that occur during apoptosis, which is a programmed cell death process [212,224]. One of the most prominent features of the nucleus in apoptotic bodies is fragmented chromatin [208]. Chromosomal DNA is cleaved into smaller fragments by CAD and EndoG during apoptosis [208,225]. These DNA fragments are tightly condensed and packaged into apoptotic bodies. In apoptosis progression, the NE undergoes structural changes, including disassembly and fragmentation [212,226]. Components of the NE, such as nuclear membrane proteins and lamins, are presented in apoptotic bodies, reflecting the disruption of NE integrity [226,227]. Notably, various nuclear proteins that are normally found within the nucleus, including transcription factors (p53; NF-κB; signal transducer and activator of transcription, STAT; cAMP response element-binding proteins, CREB proteins; forkhead box O proteins, FOXO proteins; and heat shock factor 1, HSF1), DNA-binding proteins (histone proteins including γH2AX; high-mobility group (HMG) proteins; poly(ADP-ribose) polymerase, PARP; nucleolin; Ku protein; and apoptotic endonucleases including CAD and EndoG), and chromatin-associated protein, including DNA repair factors, are included in apoptotic bodies.

During apoptosis, the formation of apoptotic bodies is a crucial step for the efficient and controlled disposal of cellular debris without inducing inflammation [212]. Effector caspases, particularly caspase-6 and caspase-3, target nuclear lamins [228,229]. Caspase-mediated cleavage of lamins A and C contributes to the disintegration of the nuclear lamina, a scaffold-like structure underlying the inner nuclear membrane [230]. The loss of nuclear envelope integrity by lamin cleavage allows the nuclear membrane to become permeable and flexible. This change in the hardness and softness may affect the equilibrium, resulting in small vesicle formation by collapse of the tension. The nuclear components, including fragmented DNA and other nuclear proteins, are encapsulated within these apoptotic bodies. Apoptotic nuclear components are initially located within the nucleus of the dying cell [212,221]. As the apoptotic process progresses, the cell undergoes membrane blebbing, which includes the formation of membranous protrusions on the cell surface and the NE [8,212]. These blebs encapsulate nuclear material, including fragmented DNA, as they pinch off from the dying cell. This process involves the caspase-mediated cleavage of nuclear lamins. Caspase-6, one of the effector caspases activated during the execution phase of apoptosis, is primarily responsible for the cleavage of nuclear lamins [230]. Caspase-6 targets lamins A and C. In some cases, while caspase-3 is not the primary caspase responsible for lamin cleavage, it can also cleave lamins, albeit to a lesser extent than caspase-6. Therefore, caspase-6 and, to some extent, caspase-3 mediate the cleavage of nuclear lamins, leading to the breakdown of the nuclear lamina and the disintegration of the NE [228,230]. This disruption of NE integrity is a critical step in the process of apoptotic body formation, as it allows nuclear material, including fragmented DNA, to mix with the cytoplasmic contents of the dying cell and become encapsulated within apoptotic bodies during membrane blebbing and pinch-off. The loss of NE integrity allows the nuclear material to mix with the cytoplasmic content of the cell. In apoptosis processes, the DNA within the nucleus becomes highly condensed and fragmented by hyperacetylation and phosphorylation of histone proteins and the increase in DNA accessibility for cleavage by endonucleases, such as CAD [231]. The condensed and fragmented DNA is more likely to be retained within the apoptotic bodies as they form. Apoptotic bodies are surrounded by a phospholipid bilayer membrane derived from the plasma membrane of the dying cell. This membrane encapsulates the contents of the apoptotic body, including nuclear material [208,212,224]. The encapsulation may prevent the release of potentially immunostimulatory nuclear material into the extracellular environment. We summarized the characteristics of cell death processes in apoptosis, necroptosis, and novel regulated cell death (RCD), such as karyoptosis (Figure 3).

## 6. Perspectives and Future Directions

The role of type II membrane proteins, which possess transcription factor activity, in maintaining nuclear membrane integrity remains a largely unexplored area. It is crucial to comprehend how these proteins contribute to the structural stability of the NE and their impact on gene regulation. Mechanistic studies should encompass the identification of key proteins, post-translational modifications, and signaling pathways that participate in maintaining the structural integrity of the NE. Subsequent research endeavors should prioritize unraveling the specific functions of type II integral proteins concerning nuclear membrane integrity. Gaining insight into how these mechanisms become dysregulated in diseases like muscular dystrophies and premature aging syndromes can offer therapeutic avenues. Furthermore, leveraging advancements in imaging technologies, such as super-resolution microscopy and live-cell imaging, can provide a more profound understanding of nuclear membrane dynamics and integrity. Investigating real-time interactions among nuclear components and their responses to various cellular cues will yield valuable insights into the preservation of nuclear morphology. To expedite progress in elucidating the molecular mechanisms of nuclear membrane integrity regulation, collaborative efforts involving cell biologists, geneticists, bioinformaticians, and clinicians are imperative. Such interdisciplinary collaborations can lead to a comprehensive comprehension of nuclear membrane integrity. The integration of data from diverse fields will offer a holistic perspective on the molecular mechanisms and clinical relevance of NE regulation. By pursuing these perspectives and future directions, researchers can advance our knowledge of nuclear membrane integrity and its significance in cellular function and disease. Ultimately, this collaborative approach holds the potential to contribute to the development of innovative treatments and therapies.

## Figures and Tables

**Figure 1 ijms-24-15497-f001:**
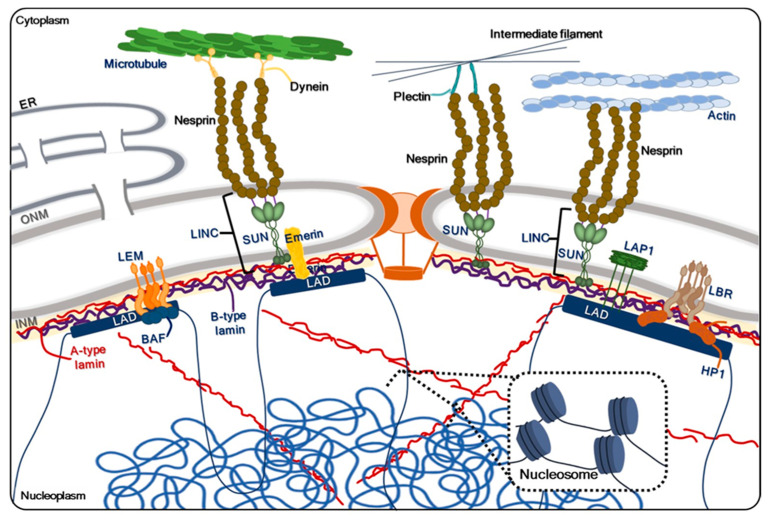
Nuclear membrane architecture and components. The NE comprises a double membrane, INM and ONM. While the ONM is a specialized extension of the ER, the INM, positioned on the nuclear side, forms a mesh network composed of nuclear lamins that serve as a structural framework. Integral membrane proteins within the INM provide mechanical support, maintaining nuclear morphology by regulating the balance of forces. This equilibrium involves counteracting the outward expansive force exerted by densely packed DNA within the nucleus and the inward restraining force of the nuclear membrane. Additionally, the outward-directed tension is reinforced by the cytoskeletal elements, contributing to the extension of the nuclear shape. To sustain this force equilibrium, integral proteins residing in the INM, including LEM, LAP, emerin, SUN, HP1, and lamin B, interact either directly or indirectly with chromatin DNA through specialized regions known as LAD via the indirect interaction. These proteins are closely associated with both the nuclear lamina and the BAF. In the cytoplasm, interactions between nesprins and actin filaments, intermediate filaments, or microtubules provide further support for the outward-directed pulling force, ultimately determining the nuclear positioning and maintaining proper nuclear shape.

**Figure 2 ijms-24-15497-f002:**
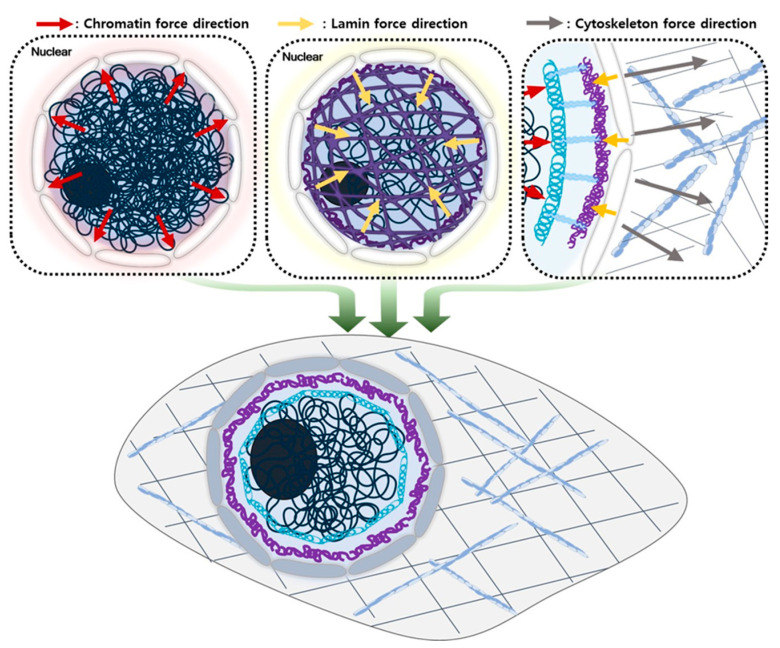
Illustration concept of force equilibrium for nuclear morphology maintenance within the nucleus. Inside the nucleus, the densely packed DNA causes it to exert an outward expansion force. To counteract this force and prevent nuclear rupture, lamin proteins form a reinforcing mesh network on the inner side of the INM, known as the nuclear lamina. Chromatin is anchored to this mesh network through integral membrane proteins, effectively serving as tension wires. These anchored chromatin regions exert an inward fastening force, reinforcing the lamina’s structural integrity, resulting in the complementation of the lamina’s resistance against the nuclear expansion force. In the cytoplasm, actin filaments, intermediate filaments, and microtubules are linked to the nuclear membrane via membrane-spanning proteins. These cytoskeletal elements play a crucial role in determining the nucleus’s position within the cell and provide the necessary force to pull the nuclear membrane outward, ensuring its proper shape is maintained.

**Figure 3 ijms-24-15497-f003:**
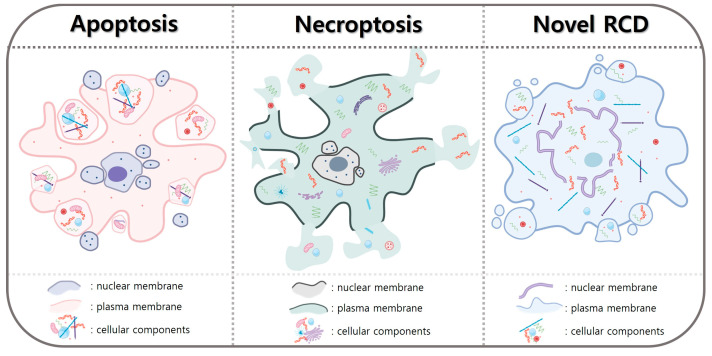
Cell death processes can induce a wide array of morphological alterations within the nuclear membrane. Apoptosis and necroptosis are two prominent forms of regulated cell death characterized by significant morphological changes in both the nuclear and cellular membranes. In apoptosis, the internal organelles within the nucleus and cytoplasm undergo encapsulation within vesicles enclosed by membranes. Consequently, it acts as a deterrent against unnecessary inflammatory responses. Conversely, necroptosis entails the formation of substantial pores in the cell membrane. Consequently, the nuclear membrane experiences rupture, leading to the direct exposure of nuclear and cellular contents to the extracellular matrix. This event triggers the release of damage-associated molecular pattern signals to neighboring cells, thus provoking an excessive inflammatory response that can give rise to inflammatory diseases.

**Table 1 ijms-24-15497-t001:** The role of each constituent involved in nuclear morphology maintenance.

Constituents	Localization	Role for the Nuclear Morphology Maintenance
Euchromatin	Inside of nucleus	Provide the force to pull the nuclear membrane towards the nucleolus and speckle regions.
Heterochromatin	Juxtaposed inside of the INM	Working in conjunction with lamina. Offer binding sites for SUN via LAD, ensuring the fixation of heterochromatin to the NE by interaction with Lamin A/C. Bind to Lamin B, resulting in anchoring chromatin to the NE. Provide an inward-pulling force against the expansive force exerted by nuclear DNA. Counteract the expansion force of packed DNA, preventing the nuclear membrane from rupturing.
Lamin A/C	Juxtaposed inside of the INM	Interacting with heterochromatin. Acting as the reinforcing bars inside the NE. Forms nuclear lamina mesh network.
Lamin B	Juxtaposed inside of INM	Interaction of heterochromatin and anchoring heterochromatin into the INM.
LEM	Juxtaposed inside of the INM	Provide binding sites for specific DNA sequences in heterochromatin to attach to the NE, anchoring chromatin in place.
LBR	Juxtaposed inside of the INM	Facilitate the interaction between post-translationally modified histones and heterochromatin, assisting in the positioning of heterochromatin at the INM.
LAD	Specific Genomic region	Provide binding sites for SUN. Anchoring heterochromatin to the NE. Recognized by lamin and CTCF, which play a role in forming heterochromatin. Characterized by strong H3K4 methylation.
SUN	Integral protein in the INM	Bind to heterochromatin’s LAD. Formation of LINC complex by interacting with proteins like nesprin in the perinuclear space. Provide binding sites for heterochromatin to attach to the NE.
Nesprin	Integral protein in the ONM	Connect SUN with cytoskeletons, including actin, intermediate filaments, or microtubules.
Plectin	Cytosol	Mediate the interaction between nesprin and intermediate filaments at the cytoplasm.
Dynein	Cytosol	Facilitate the interaction between nesprin and microtubules at the cytoplasm.
Actin, Intermediate filaments ,Microtubules	Cytosol	Interaction with nesprin, plectin, and Dynein Determine the position of the nucleus within the cell Assist to pull the nuclear membrane outward to maintain the normal nuclear shape.

## Data Availability

Not applicable.

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
