# Peer review of "Molecular Mechanisms for the Regulation of Nuclear Membrane Integrity"

_ijms, 2023, doi:10.3390/ijms242015497_

Round 1
Reviewer 1 Report
Molecular mechanisms for the regulation of inner nuclear membrane integrity
In the above review authors summarise our current knowledge of the constituents (proteins) of nuclear envelope (NE), important protein constituents of the INM, their roles and NE-associated pathologies. Although, there are similar reviews on the subject the manuscript will be complementary to them. The manuscript is well written with a summary table of the relevant proteins and summary figures. Addressing following suggestions may improve the quality of the manuscript.
Major Comments
1. Although, half the review focuses on the INM, rest of the review covers all aspects of the nuclear envelope including outer membrane, perinuclear spaces and proteins, ATF6 and nuclear membrane rupture (apoptosis) etc. Therefore, it will be appropriate to modify the title of the review to be more generic to reflect the contents.
2. With many thousands of papers and a couple of thousand reviews on the nuclear envelope (structure, function, remodeling, and pathologies) it is hard to cite all of them. However, some relevant recent/classic reviews on the subject (Nuclear envelope=NE) not cited: e.g. PMID: 37660480 (2023, NE dynamics and disease), PMID: 35757775 (2022, NE and immunity), PMID: 33421755 (2021, NE remodeling during mitosis), PMID: 32530796 (2020, NE identity), PMID: 32291910 (2020, NE and ageing), PMID: 32109733 (2020, NE and meiosis), PMID: 28120913 (2017, NE remodeling), PMID: 27389815 (2016, NE myopathies), PMID: 26437591 (2016, QC and NE).
3. Figure captions: All three figure captions are very long and does not directly describe the figure. Figure captions should be concise and provide only necessary and sufficient details to understand the figure such as the descriptions of the structures not their functions or pathologies etc. Move all the other detail descriptions to the text.
4. Chromatin, Heterochromatin, Euchromatin, and INM: Chromatins are not integral constituents of INM per se and describing them as such is incorrect. They are not technically proteins, rather nucleoprotein complexes and it is incorrect to list them in the Table 1 as the proteins of the INM and describe them as such in section 4 for as constituents of INM. This goes true for HP1 as well. Remove this subsection. However, INM plays a crucial role in the chromatin states and therefore bring them up in relevant subsections briefly.
5. Figure 3: Needs to label the structures shown in the three images. Readers have no idea where the cell membrane or NE etc are. RCD (regulated cell death) not abbreviated any where in the figure caption or text. Also see comment 3 above.
Minor Comments
1. Inner Nuclear Membrane (INM) and Outer Nuclear Membrane (ONM) were already abbreviated in the introduction (Lines 39-40) and the abbreviations doesn’t need to be introduced again further down (Lines 58-59).
2. Table 1: Needs to have a title which describes the content
3. Table 1: It seems it was submitted as an image and therefore the resolution is very poor. Please provide as a table (If not possible as a high-resolution image).
Author Response
Oct-18, 2023
Jim Wang, Section Managing Editor
International Journal of Molecular Sciences
RE: ijms-2658462R1
(Molecular mechanisms for the regulation of inner nuclear membrane integrity)
Dear Editor:
With this letter, we are submitting our revised manuscript (ijms-2658462R1) titled "Molecular Mechanisms for the Regulation of Nuclear Membrane Integrity" by Lee et al. for your consideration for publication in the International Journal of Molecular Sciences. This work has not been previously published and is not under consideration for publication elsewhere. The authors declare no conflicts of interest that might have influenced the results or content of the manuscript.
In the revised manuscript, we have made additions, reorganized sections, and addressed the reviewer's comments point by point. The revised sections are highlighted for easy identification. Consequently, we believe that the revised review article now provides stronger support for understanding the molecular mechanisms regulating nuclear membrane integrity. The content of the review offers valuable insights into nuclear envelope structure and function, with implications for cell biology and cancer research.
Reviewer 1
Molecular mechanisms for the regulation of inner nuclear membrane integrity
General comment. In the above review authors summarise our current knowledge of the constituents (proteins) of nuclear envelope (NE), important protein constituents of the INM, their roles and NE-associated pathologies. Although, there are similar reviews on the subject the manuscript will be complementary to them. The manuscript is well written with a summary table of the relevant proteins and summary figures. Addressing following suggestions may improve the quality of the manuscript.
Response for general comment. We are grateful to the reviewer for recognizing the importance of our work. We have diligently addressed each of the reviewer's comments in a point-by-point manner and made the necessary corrections in the manuscript. These revisions have been clearly indicated in the corrected sections of the revised manuscript.
Major Comments
Comment 1. Although, half the review focuses on the INM, rest of the review covers all aspects of the nuclear envelope including outer membrane, perinuclear spaces and proteins, ATF6 and nuclear membrane rupture (apoptosis) etc. Therefore, it will be appropriate to modify the title of the review to be more generic to reflect the contents.
Response 1. We appreciate the constructive comment from the reviewer. We agree with the reviewer's opinion. Therefore, we have changed the title of the manuscript to "Molecular Mechanisms for the Regulation of Nuclear Membrane Integrity."
Comment 2. With many thousands of papers and a couple of thousand reviews on the nuclear envelope (structure, function, remodeling, and pathologies) it is hard to cite all of them. However, some relevant recent/classic reviews on the subject (Nuclear envelope=NE) not cited: e.g. PMID: 37660480 (2023, NE dynamics and disease), PMID: 35757775 (2022, NE and immunity), PMID: 33421755 (2021, NE remodeling during mitosis), PMID: 32530796 (2020, NE identity), PMID: 32291910 (2020, NE and ageing), PMID: 32109733 (2020, NE and meiosis), PMID: 28120913 (2017, NE remodeling), PMID: 27389815 (2016, NE myopathies), PMID: 26437591 (2016, QC and NE).
Response 2. We appreciate the reviewer's comment and suggestion. We have incorporated the relevant information based on the suggested literature with proper citation. Please refer to the revised manuscript for details.
Comment 3. Figure captions: All three figure captions are very long and does not directly describe the figure. Figure captions should be concise and provide only necessary and sufficient details to understand the figure such as the descriptions of the structures not their functions or pathologies etc. Move all the other detail descriptions to the text.
Response 3. We appreciate the reviewer's constructive and invaluable comments. We concur with the reviewer's opinion. In order to avoid redundancy, we have made the figure captions more concise. Additionally, we have relocated the detailed explanations for the figures to the main text in the revised manuscript.
Comment 4. Chromatin, Heterochromatin, Euchromatin, and INM: Chromatins are not integral constituents of INM per se and describing them as such is incorrect. They are not technically proteins, rather nucleoprotein complexes and it is incorrect to list them in the Table 1 as the proteins of the INM and describe them as such in section 4 for as constituents of INM. This goes true for HP1 as well. Remove this subsection. However, INM plays a crucial role in the chromatin states and therefore bring them up in relevant subsections briefly.
Response 4. We thank the reviewer for the comment. We also apologize for any oversights. In response to the suggestion, we have retitled section 4 as "The Roles of constituents in the NE involved in nuclear morphology maintenance by complex formation." We have also changed the title of Table 1 to "The Role of each constituent in the NE involved in nuclear morphology maintenance by complex formation." In Table 1, the first column heading "Protein" has been changed to "Constituents.". Additionally, following the reviewer's suggestion, we have reorganized section 4 into two subsections: "4.1 Constituents tethered in the INM" and "4.2 Integral membrane proteins in the INM," in the revised manuscript.
Comment 5. Figure 3: Needs to label the structures shown in the three images. Readers have no idea where the cell membrane or NE etc are. RCD (regulated cell death) not abbreviated any where in the figure caption or text. Also see comment 3 above.
Response 5. We appreciate the reviewer's helpful feedback. We concur with the suggestions provided by the reviewer. In line with the reviewer's recommendations, we have included illustrations to indicate the structures in the three images. Furthermore, we have introduced a general concept for regulated cell death derived from abnormal nuclear morphology changes at the beginning of section 5 in the revised manuscript.
Minor Comments
Minor comment 1. Inner Nuclear Membrane (INM) and Outer Nuclear Membrane (ONM) were already abbreviated in the introduction (Lines 39-40) and the abbreviations doesn’t need to be introduced again further down (Lines 58-59).
Response for the minor comment 1. We thoroughly reviewed the entire manuscript and made corrections to all abbreviations in the revised version.
Minor comment 2. Table 1: Needs to have a title which describes the content.
Response for the minor comment 2. We apologize for the oversight. In the revised manuscript, we have added an appropriate title to Table 1.
Minor comment 3. Table 1: It seems it was submitted as an image and therefore the resolution is very poor. Please provide as a table (If not possible as a high-resolution image).
Response for the minor comment 3. We apologize for the oversight. In the revised manuscript, we have replaced Table 1 with a higher resolution table.
Reviewer 2
General comment. In this review article, the authors intended to describe mechanisms that regulate the integrity of the nuclear membrane. However, due to problems in sections 4 and 5, I find it challenging to get what the authors tried to show in this review article.
Response to General comment. We appreciate the reviewer's recognition of the importance of our review article and their constructive comments, particularly regarding sections 4 and 5. In response to the feedback from reviewer 1, we have re-organized and clarified the challenges associated with type II membrane proteins in the INM.
Major Comments
Comment 1. The title of section 4 is not appropriate. Although the title of this section is “Constituents in the INM involved in nuclear morphology maintenance,” the authors included descriptions of chromatins and HP1. This is confusing. I recommend that the authors focus on the maintenance of nuclear morphology in this section. Descriptions of components of the INM, such as SUN, LBR, and lamins, can be included in section 2.2.
Response 1. We appreciate the reviewer's constructive and invaluable comments. In section 2.2, we provided a simple explanation of the proteins as architectural components in the INM. However, in section 4, we expanded on how these proteins are involved in maintaining nuclear membrane integrity. To avoid any confusion and in accordance with the feedback from reviewer 1 and 2, we have re-organized section 4 into two categories: "4.1. Constituents tethered in the INM" and "4.2. Integral membrane proteins in the INM." Additionally, we have added further explanations to clarify how these proteins play a role in maintaining nuclear membrane integrity in the revised manuscript.
Comment 2. Please revise Table 1. The title and legend of this table are missing. In line 225, the authors described that “Table 1 summarized the role of each INM integral protein”. However, this table has proteins in the other locations, such as nesprins and Plectin. Chromatins and LAD are not proteins of INM. Heterochromatin appeared four times in this table. What are the differences?
Response 2. We apologize for our oversight and appreciate the reviewer's feedback. We agree with the reviewer's suggestion, and similar issues were pointed out by reviewer 1 as well. To enhance clarity, we have revised the title of section 4 to "The Roles of constituents in the NE involved in nuclear morphology maintenance by complex formation". Furthermore, we have undertaken an extensive reorganization of section 4 to improve the clarity of this review article. Additionally, we have added a title for Table 1 in the revised manuscript.
Comment 3. LBR should be included in the section on integral membrane proteins. The description of MAN1 needs to be added in section 4.
Response 3. We appreciate the reviewer's invaluable comment. Following the reviewer's suggestion, we have relocated LBR to the section on integral membrane proteins. Furthermore, we have added MAN1 in a new subsection, 4.2.4, under section 4.2, titled "Integral membrane proteins in the INM".
Comment 4. ATF6 is not a good example of a transcription factor in INM. Localization of ATF6 is not restricted to the INM. Mechanisms of ATF6 activation described in this section are not related to the INM.
Response 4. We appreciate the reviewer's understanding. One of the objectives of this review paper is to introduce our current ongoing study. As mentioned in the ATF6 subsection of section 4.2.5, there has been limited attention paid to the localization of these type II membrane transcription factors. Our manuscript, which is currently under review, explores how aberrant regulation of these type II membrane proteins at the INM can induce a new type of cell death. Therefore, we included this information in the review to provide context for our ongoing research.
Comment 5. The title of section 5 is not appropriate. There is no description of the release of tethered DNA in this section. What is the release of tethered DNA from the INM? What is the significance of this phenomenon?
Response 5. We appreciate the invaluable comments from the reviewer and agree with their opinion. We have made a change to the title, which now reads "Nuclear membrane breakdown and nuclear morphology change". Additionally, we have introduced a general explanation of the significance of nuclear morphology changes. Furthermore, we have expanded on the information concerning the dynamic nature of the nuclear envelope in the revised manuscript.
Comment 6. In section 5.1., the authors described cleavage of SUN and KASH proteins by protease. However, the relation of the proteolysis of these proteins to the release of tethered DNA is not described.
Response 6. We appreciate the constructive comment from the reviewer. As described in the revised manuscript in Section 5, nuclear morphology changes are triggered by the disruption of the tension equilibrium between outward expansion and inward fastening forces. Given that the maintenance of nuclear shape is directly linked to nuclear membrane integrity homeostasis, nuclear membrane anchoring proteins and type II membrane proteins may play an essential role in regulating nuclear membrane integrity. In this context, protease-mediated cleavage of SUN and KASH proteins and the potential involvement of ER or Golgi resident proteases may be crucial for maintaining nuclear shape through the regulation of nuclear membrane integrity, especially when cells are exposed to various stresses, such as unfolded protein stress in the ER. We have extensively reorganized and modified the content to better focus on the objective of this review in the revised manuscript.
Comment 7. The title of section 5.2. is “Proteases in the perinuclear space.” However, the authors focused on S1P and S2P proteases, which function in regulated intramembrane proteolysis. Descriptions of the functions of S1P and S2P proteases in the nuclear envelope are mostly speculation.
Response 7. We appreciate the reviewer's invaluable comment, and we concur with their opinion. The content in the previous Section 5.2 was primarily based on our ongoing experimental results, which have not been published as of yet. Consequently, we acknowledge that this section contained a substantial amount of speculation. To avoid potential confusion, we have decided to omit the previous Section 5.2 from the revised manuscript.
Comment 8. The authors described nuclear blebbing in the title of section 5.3. However, the contents of this section are nuclear membrane breakdown and nuclear membrane rupture. Nuclear blebbing was not well described in this section.
Response 8. We appreciate the reviewer's constructive comment, and we have made revisions accordingly. While we extensively reviewed the existing literature, it is important to note that the molecular mechanisms underlying the processes of blebbing and material encapsulation within the nuclear envelope have not been definitively elucidated. We acknowledge that this represents an intriguing research avenue for future investigations in the field.
Comment 9. Relations between nuclear membrane integrity and apoptosis should be explained in more detail.
Response 9. Thank you for your suggestion. In order to provide accurate and clear information, we have rewritten and reorganized both sections 4 and 5 in the revised manuscript.
Minor comments
Minor comment 1. Line 133: The authors defined PNS as an abbreviation for perinuclear space. However, this abbreviation is not used in this manuscript.
Minor response 1. Thank you for pointing that out. In the revised manuscript, we have spelled out the abbreviation "PNS" to provide clarity to the readers.
Minor comment 2. Line 388: What does CTCF stand for?
Minor response 2. In the revised manuscript, you have spelled out the abbreviation "CTCF" as "CCCTC-binding factor" to ensure clarity for readers.
Minor comment 3. Definitions of CAD and EndoG appeared twice (lines 676 and 697).
Minor response 3. We apologize for our oversight. We have deleted and corrected this sentence in the revised manuscript.
Thus, we hope that you will agree that we have addressed each reviewer’s comments adequately and that this manuscript can now be considered for publication in International Journal of Molecular Sciences.
Sincerely,
Yong-Yeon Cho, Ph. D., Professor, Dean
College of Pharmacy, The Catholic University of Korea
43, Jibong-ro, Wonmi-gu, Bucheon-si, Gyeonggi-do 420-743, Republic of Korea
Phone: +82-2-2164-4092
Fax: +82-2-2164-4059
E-mail: yongyeon@catholic.ac.kr, choyycho@gmail.com

Reviewer 2 Report
In this review article, the authors intended to describe mechanisms that regulate the integrity of the nuclear membrane. However, due to problems in sections 4 and 5, I find it challenging to get what the authors tried to show in this review article.
Major points
1. The title of section 4 is not appropriate. Although the title of this section is “Constituents in the INM involved in nuclear morphology maintenance,” the authors included descriptions of chromatins and HP1. This is confusing. I recommend that the authors focus on the maintenance of nuclear morphology in this section. Descriptions of components of the INM, such as SUN, LBR, and lamins, can be included in section 2.2.
2. Please revise Table 1. The title and legend of this table are missing. In line 225, the authors described that “Table 1 summarized the role of each INM integral protein”. However, this table has proteins in the other locations, such as nesprins and Plectin. Chromatins and LAD are not proteins of INM. Heterochromatin appeared four times in this table. What are the differences?
3. LBR should be included in the section on integral membrane proteins. The description of MAN1 needs to be added in section 4.
4. ATF6 is not a good example of a transcription factor in INM. Localization of ATF6 is not restricted to the INM. Mechanisms of ATF6 activation described in this section are not related to the INM.
5. The title of section 5 is not appropriate. There is no description of the release of tethered DNA in this section. What is the release of tethered DNA from the INM? What is the significance of this phenomenon?
6. In section 5.1., the authors described cleavage of SUN and KASH proteins by protease. However, the relation of the proteolysis of these proteins to the release of tethered DNA is not described.
7. The title of section 5.2. is “Proteases in the perinuclear space.” However, the authors focused on S1P and S2P proteases, which function in regulated intramembrane proteolysis. Descriptions of the functions of S1P and S2P proteases in the nuclear envelope are mostly speculation.
8. The authors described nuclear blebbing in the title of section 5.3. However, the contents of this section are nuclear membrane breakdown and nuclear membrane rupture. Nuclear blebbing was not well described in this section.
9. Relations between nuclear membrane integrity and apoptosis should be explained in more detail.
Specific points
1. Line 133: The authors defined PNS as an abbreviation for perinuclear space. However, this abbreviation is not used in this manuscript.
2. Line 388: What does CTCF stand for?
3. Definitions of CAD and EndoG appeared twice (lines 676 and 697).
Author Response

(The authors gave the same response as above.)

Round 2
Reviewer 2 Report
In this revised manuscript, the authors reorganized the manuscript according to the reviewers’ suggestions. I think the manuscript was much improved. Comments to the revised manuscript are as follows.
1. I think Table 1 still needs to be revised. Please remove the phrase “in the NE” from the title since this table has proteins in the cytosol. It is not well described why SUN and heterochromatin appeared twice and four times, respectively, in this table. I do not understand the meaning of the “Binding partner for complex formation”. For example, in the upper low of SUN, the authors describe that SUN binds to LAD in the “Roles” column but its binding partner is “-“. In the lower low of SUN, this protein is shown to bind to LAD. What is the difference? The definition of LINC should be moved to the first appearance in the text. The phrase “intermediated filaments” should be “intermediate filaments”.
2. line 124: Remove “(PNS)” from the text. This abbreviation is not used in this manuscript.
3. The definition of the SUN domain is found five times (lines 123, 418, 419, 425 and within Table 1).
4. line 537: What is Gc?
5. line 556: ATG6-GFP should be ATF6-GFP.
6. line 625: “homeosis” Did the authors mean to write “homeostasis”?
7. Section 5.3: Please use the phrase “nuclear blebbing” in the first paragraph of this section.
8. line 755: Remove “(DAMP)” from the text. This abbreviation is not used in this manuscript.
Author Response
Oct-20, 2023
Jim Wang, Section Managing Editor
International Journal of Molecular Sciences
RE: ijms-2658462R2
(Molecular mechanisms for the regulation of nuclear membrane integrity)
Dear Editor
With this letter, we are submitting our revised manuscript (ijms-2658462R2) titled "Molecular Mechanisms for the Regulation of Nuclear Membrane Integrity" by Lee et al. for your consideration for publication in the International Journal of Molecular Sciences.
In the revised manuscript, we have made several additions to enhance the clarity of the review article and have removed redundancies in the use of abbreviations, in accordance with the invaluable and constructive comments provided by the reviewer. We have carefully addressed each of the reviewer's comments in a point-by-point manner, as outlined below.
Reviewer 2
Overall comment. In this revised manuscript, the authors reorganized the manuscript according to the reviewers’ suggestions. I think the manuscript was much improved. Comments to the revised manuscript are as follows.
Response to overall comment. We appreciate the reviewer's acknowledgment. We have implemented changes based on the reviewer's constructive comments. Please review our revised manuscript
Comment 1. I think Table 1 still needs to be revised. Please remove the phrase “in the NE” from the title since this table has proteins in the cytosol. It is not well described why SUN and heterochromatin appeared twice and four times, respectively, in this table. I do not understand the meaning of the “Binding partner for complex formation”. For example, in the upper low of SUN, the authors describe that SUN binds to LAD in the “Roles” column but its binding partner is “-“. In the lower low of SUN, this protein is shown to bind to LAD. What is the difference? The definition of LINC should be moved to the first appearance in the text. The phrase “intermediated filaments” should be “intermediate filaments”.
Response 1. We apologize for the confusing format of Table 1. We have reformatted Table 1 without disrupting the flow of the main text. Additionally, we have removed "in the NE" from the table title.
Comment 2. line 124: Remove “(PNS)” from the text. This abbreviation is not used in this manuscript.
Response 2. We appreciate the reviewer's comment, and we have removed the abbreviation in the revised manuscript.
Comment 3. The definition of the SUN domain is found five times (lines 123, 418, 419, 425 and within Table 1).
Response 3. We apologize for our inattention. We have removed repetitive definitions of SUN in the revised manuscript.
Comment 4. line 537: What is Gc?
Response 4. We apologize for our inattention. "Gc" stands for the Golgi complex, and we have appropriately abbreviated it as "Gc" in the revised manuscript.
Comment 5. line 556: ATG6-GFP should be ATF6-GFP.
Response 5. We appreciate the reviewer's attention to detail. The reviewer is correct, and we have made the change to "ATF6-GFP" in the revised manuscript.
Comment 6. line 625: “homeosis” Did the authors mean to write “homeostasis”?
Response 6. We appreciate the reviewer's comment, and we have made the change to "homeostasis" in the revised manuscript.
Comment 7. Section 5.3: Please use the phrase “nuclear blebbing” in the first paragraph of this section.
Response 7. We appreciate the reviewer's constructive comment. In response to this, we have added a sentence regarding nuclear blebbing in the first paragraph of section 5.3 in the revised manuscript.
Comment 8. line 755: Remove “(DAMP)” from the text. This abbreviation is not used in this manuscript.
Response 8. We appreciate the reviewer's constructive comment. In light of this, we have removed the abbreviation "DAMP" from the Figure 3 legend in the revised manuscript.
Thus, we hope that you will agree that we have addressed each reviewer’s comments adequately and that this manuscript can now be considered for publication in International Journal of Molecular Sciences.
Sincerely,
Yong-Yeon Cho, Ph. D., Professor, Dean
College of Pharmacy, The Catholic University of Korea
43, Jibong-ro, Wonmi-gu, Bucheon-si, Gyeonggi-do 420-743, Republic of Korea
Phone: +82-2-2164-4092
Fax: +82-2-2164-4059
E-mail: yongyeon@catholic.ac.kr, choyycho@gmail.com
